

# Spatial variation of parrotfish assemblages at oceanic islands in the western Caribbean: evidence of indirect effects of fishing?

Natalia Rivas[1], Arturo Acero P.[1] and José Tavera[2]

[1] Instituto de Estudios en Ciencias del Mar (Cecimar), Universidad Nacional de Colombia sede Caribe, El Rodadero, Santa Marta, Colombia
[2] Departamento de Biología, Universidad del Valle, Cali, Colombia

## ABSTRACT

Fish populations that bear considerable pressure levels tend to show a decline in the average size of individuals, with the small and unexploited species replacing the large and exploited ones. It is important to carry on with their characterization in areas where they are becoming an important source of food for local human populations. An example of such species are parrotfishes, whose responses to external factors such as fishing need to be understood and predicted. In this study, we used a diver-operated stereo-video to examine individual body size, sex ratios and proportion of species of the parrotfish assemblage and analyze them on a qualitative fishing pressure gradient at four oceanic islands in the Colombian Caribbean. We reported over 10,000 occurrences of eleven parrotfish species, of which we estimated the total length of over 90%, grouping them into three size categories (large, medium, and small). Our data showed a spatial variation of parrotfishes' abundances, biomass, and individual body size. Observed differences are size-category-dependent throughout the qualitative fishing pressure. In general, the medium-bodied species had smaller sizes, lower abundances, and thus lower contribution to the total parrotfish biomass at the most heavily fished island. Unexpectedly, we found evidence of possible indirect effects over the small-bodied species *Scarus iseri* and *Scarus taeniopterus* with significantly greater abundances, and larger sizes of males of *S. iseri*, at the higher fishing pressure sites. Overall, our data highlights the extent of the spatial variation in the parrotfish communities at relatively short distances, and present new insights into the responses of parrotfish species on a spectrum of body sizes along a gradient of human pressure.

# INTRODUCTION

Local stressors such as overfishing have caused significant changes in the structure and function of coral reefs (*Jackson et al., 2001*; *McLean et al., 2016*). Overexploited fisheries tend to shift towards lower trophic levels (*Taylor, 2014*), with growing captures of smaller and less commercial species, usually herbivores (*Burke et al., 2011*). The assessment of

Corresponding author
Natalia Rivas, narivase@unal.edu.co

herbivores populations in coral reefs subject to fishing activities is compulsory, yet the lack of rigorous data on fishing effort remains one of the main obstacles to understanding fishing effects (*Stewart et al., 2010*). Therefore, to assess populations in localities with poorly documented coastal fishing efforts, it is necessary to resort to additional information linked to such efforts.

Different measures, such as human population density, have been proposed to indicate fishing effort (*Stewart et al., 2010*). There are two ways in which human density can affect fish populations locally (*Williams et al., 2008*). Directly, through overfishing, with a selective effect on targeted species, and indirectly, through habitat degradation, with a more widespread effect on both targeted and non-targeted species (*Pauly, 1990*; *Williams et al., 2008*; *Stewart et al., 2010*). Studies in insular territories around the world have shown a negative relationship between human population density and biomass and abundance of fish populations in coral reefs adjacent to inhabited territories (*Clua & Legendre, 2008*; *Williams et al., 2008*; *Bellwood, Hoey & Hughes, 2012*). Related information such as access to local markets and urbanization are also reliable indicators to approach the fishing pressure level of a particular locality (*Aswani & Sabetian, 2009*; *Brewer et al., 2012*, *2013*; *Cinner et al., 2013*). In addition, different biological attributes in fish populations are also proposed as features to study fishing effects.

Individual body size is one of the most operational features to evaluate the effects of overfishing (*Rochet & Trenkel, 2003*). However, size-based analyses have typically examined assemblage-level patterns of the entire reef fish communities (*Jennings et al., 2002*; *Dulvy et al., 2004*; *Graham et al., 2005*), ignoring possible variation within groups of fishes. In the Caribbean, size-based studies have shown a turnover of fisheries towards species of lower trophic levels, such as parrotfishes (*Hawkins & Roberts, 2004*; *Hardt, 2009*; *Vallès & Oxenford, 2014*) and a size-based study conducted on this group can add resolution to the existing literature on this topic.

Parrotfishes (Labridae: Scarinae) and surgeon fishes (Acanthuridae) are two of the main herbivore clades in coral reefs of the Caribbean (*Mumby et al., 2006*). Former studies in the Caribbean have linked patterns of spatial variation in parrotfish communities with overfishing. Those studies found a negative relationship between fishing pressure and biomass of large-bodied species and with the individual body weight of all parrotfish species (*Hawkins & Roberts, 2004*; *Vallès & Oxenford, 2014*; *Vallès, Gill & Oxenford, 2015*). Overfishing has also been associated with indirect effects on fish communities through compensatory increases of small species when the larger ones decrease (*Dulvy et al., 2004*). However, there is scarce evidence of this effect in parrotfish communities. A largescale study by *Hawkins & Roberts (2003)* found that the abundance of the small-bodied *Sparisoma atomarium* increased with fishing pressure. *Vallès, Gill & Oxenford (2015)* found that at higher fishing pressure, *Scarus iseri* and *Scarus vetula* were still more and less abundant, suggesting a possible event of competitive release between the two species, but no significant differences supported such hypothesis. Therefore, it is relevant to provide a better characterization of parrotfish spatial variation to explore emergent exploitation impacts within fisheries, especially in small-scale spatial gradient of fishing pressure and in areas where these effects are still unknown.

Furthermore, parrotfishes are an interesting group to assess fishing effects on sex ratios and body sizes of protogynous species, the most common form of sequential hermaphroditism of reef fishes (*Gemmell et al., 2019*), an issue than remains poorly examined (*Buxton, 1993*; *Coleman, Koenig & Collins, 1996*; *McGovern et al., 1998*; *Hawkins & Roberts, 2003*; *O'Farrell et al., 2016*). In Bonaire, abundances of males of *S. vetula*, *Sparisoma viride*, *Sparisoma aurofrenatum* and *Scarus taeniopterus* increased after a fishing ban (*O'Farrell et al., 2016*). Likewise, most parrotfish species evaluated by *Hawkins & Roberts (2003)* showed a decreased in body sizes of males and females with fewer males at higher fishing pressure. In marine protected areas, the authors also found an increase in time of body sizes of males and females of most species. Hence, adding resolution to these relationships is pertinent to monitor their populations over time.

In this study, we used diver-operated stereo-video to assess parrotfish populations and explore the effects of fishing pressure in oceanic islands of the Colombian Caribbean. The stereo-video technique allowed us to gather high-resolution body-length data for a size-based analysis under a qualitative fishing pressure gradient in which we included the best available local information. We anticipated that fishing exploitation would reduce the abundance and average body size of medium and large-bodied species, with a minor contribution of those species to the overall parrotfish biomass. Consequently, small-sized species would have a greater contribution to the overall parrotfish biomass either by having greater abundances at higher fishing pressure or mainly because of the lower contribution of the medium and large-bodied species proportions.

## MATERIALS AND METHODS

### Sampling sites and fishing pressure gradient

The Seaflower Biosphere Reserve (SBR) is a complex of islands, atolls, cays, and shoals in the Colombian Caribbean, including one of the largest Caribbean coral reefs complexes (*Abril-Howard et al., 2012a*) and over 77% of the coral formations of the country (*Abril-Howard et al., 2012b*). We recorded parrotfish species in coral reef areas of several islands of the SBR: San Andrés (SA), Bolívar Cay (BOL), Albuquerque Cay (ALB), and Providencia and Santa Catalina (PRO). The main island of the archipelago is SA, located 90 km south of PRO, the other inhabited populated island, while BOL and ALB are two small uninhabited cays, 25 and 37 km south of SA, respectively (Fig. 1).

Considering the selectivity of fisheries for large fishes (*O'Farrell et al., 2016*), metrics such as individual body size, sex ratio, and overall proportions of parrotfishes would reflect different responses to exploitation by comparing them across a fishing gradient (*Rochet & Trenkel, 2003*; *Dulvy et al., 2004*). However, at the SBR, no standardized information precludes us to quantify fishing pressure. Therefore, we considered additional drivers based on the literature of the topic (*Clua & Legendre, 2008*; *Williams et al., 2008*; *Aswani & Sabetian, 2009*; *Bellwood, Hoey & Hughes, 2012*; *Brewer et al., 2012*, *2013*; *Cinner et al., 2013*). To include species with different levels of interest for the fisheries, we evaluated eleven species of parrotfish across a wide range of body sizes along the established human pressure gradient. We separated the species into three categories: small-sized ones (maximum total length <30 cm) including *S. taeniopterus*, *S. iseri*, *S. aurofrenatum* and

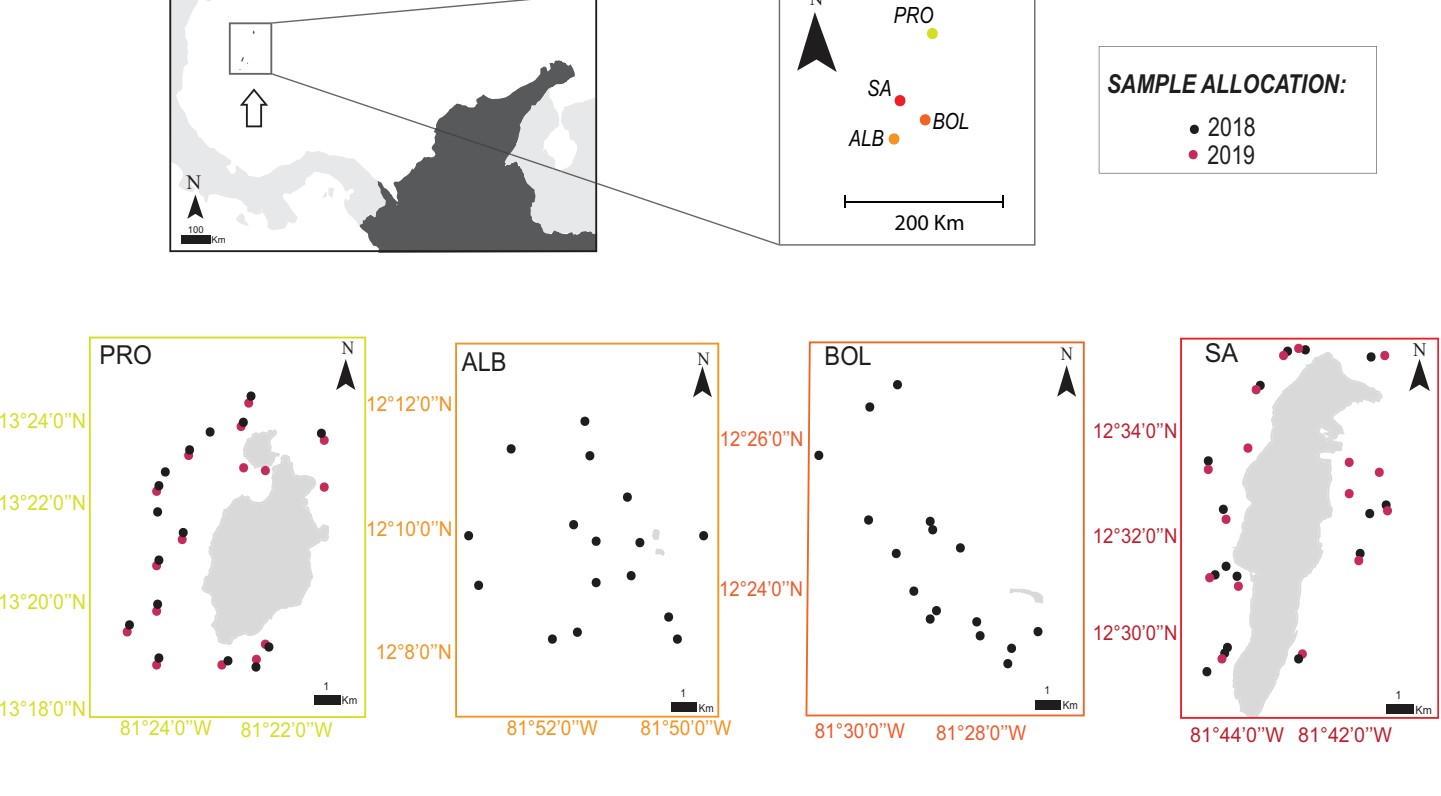

**Figure 1 Location of the four sampling sites along the Seaflower Biosphere Reserve.** Sample allocations of 2018 and 2019 surveys are specified with red and black dots. Stations evaluated at each locality are georeferenced (QGIS v.3.14). Insert map shows location of the reserve within the Colombian Caribbean (white arrow). Localities: Providencia and Santa Catalina (PRO); Albuquerque (ALB); Bolívar (BOL); San Andrés (SA).

*S. atomarium*; medium-sized species (maximum total length >30–<70 cm) *Sparisoma chrysopterum* and *Sparisoma rubripinne*, in addition to *S. vetula* and *S. viride*. The large-bodied species (maximum total length >70 cm) included *Scarus coelestinus*, *Scarus coeruleus*, and *Scarus guacamaia*.

The proposed ranking of localities from highest to lowest fishing pressure was: SA, BOL, ALB, and PRO. This order was based on: (1) Human population density (number of inhabitants per square kilometer of reef structure) in SA is 784 inhabitants/km$^2$, while in PRO is 18 inhabitants/km$^2$ (*Gamboa et al., 2012*; *DANE, 2016*; *Cámara de Comercio de San Andrés, Providencia y Santa Catalina, 2018*). Bolívar and Albuquerque cays have no human population besides a dozen Colombian soldiers; however, fishers of San Andrés frequently visit these islets due to their proximity to the main island. (2) SA has easy access to fishing grounds from the coastline, allowing for shore fishing. This activity is conducted by independent fishers, whose population is not characterized and quantified, and targets

multiple species, including parrotfishes, grunts (Haemulidae), and jacks (Carangidae), with unknown catch volumes (*Castro, 2005*). (3) In 1953, SA was designated as a tax-free region, which promoted an enormous increase in tourism and a significant immigration of people to the island (*Meisel, 2003*; *Abril-Howard et al., 2012a*). Artisanal fishery went from local subsistence to a specialized and institutionalized activity based on the local market demand (*Olmos, 2019*). In contrast, those changes have not significantly impacted PRO (*Meisel, 2003*; *Abril-Howard et al., 2012a*). (4) Catch per unit effort (CPUE) in SA has shown a decline since 2009, which has been related to an increase in overfishing on the island (*Santos-Martínez, García & Rojas, 2017*; *Santos-Martínez et al., 2019*). Additionally, reports show a greater effort and capture volumes in SA over nearby fishing grounds, including BOL and ALB cays (*Castro, Grandas & García, 2007*; *Rojas et al., 2015*). The fishing pressure distinction between BOL and ALB was based merely on their distance from SA.

## Surveys

We used a diver-operated stereo-video approach (*Harvey, Fletcher & Shortis, 2003*; *Langlois et al., 2010*) to collect field data. Before each field trip, we performed the stereo-video calibration using the CAL software (SeaGIS Pty Ltd. Bacchus Marsh, Victoria, Australia) following *Harvey & Shortis (1998)*. During the calibration process, a cube with approximately 80 targets was filmed in 20 different positions with the stereo-video system. With the CAL software, the targets were located in those different positions of the cube, generating a calibration archive with the exact configuration of the camera system, including the inward angle and distance between the cameras, among other parameters. After testing the calibration with a known-size devise, the stereo video was set to be used in the field (Fig. S1A).

The stereo-video technique allowed us to gather reliable data on individual body size, sex ratio, and overall proportions of the parrotfishes' assemblage in the SBR with high resolution and low estimation error. In the absence of previous data from this technique to compare with, four considerations influenced our final sampling design: (i) to sample each location as close together in time as possible, (ii) to select sites with similar coral reef bottoms and with a similar number of shallow (3–8 m), medium (8–15 m) and deep (15–30 m) stations surveyed at each locality (Table S1), (iii) to minimize associated researcher biases by having the same researcher handling the stereo-video and processing the videos, and (iv) to produce a balanced set of samples across errant diver specific timed surveys.

## Sample allocation and video processing

We sampled 16 stations per locality for a total of 64 stations between October and November 2018 in SA, BOL, ALB, and PRO, and 32 stations a year later in 2019 in SA and PRO (Fig. 1; Table S1). In addition, in 2019, we sampled a greater number of shallower stations (<6 m) close to seagrass meadows to gather more data on *S. rubripinne* and *S. chrysopterum*, which had a small sample size in the 2018 surveys. Consequently, the
sampling design was unbalanced when comparing SA and PRO between the 2 years (Table S1).

We used the guides of *Humann & Deloach (2014)* and *Robertson et al. (2015)* to identify the individuals to species level and to give a sex designation. Following *Hawkins & Roberts (2004)*, we considered parrotfishes in juvenile and initial color phases as females and in terminal color phases as males. In the case of *S. coelestinus*, with no color phases, there was no sex designation. To distinguish between the initial phases of *S. taeniopterus* and *S. iseri* we used the color of the edges of the caudal fin (light for the first and dark brown upper and lower edges for the second species) (*Robertson et al., 2015*). From the videos, we recorded occurrence, sex, and total length data (TL), measured from the tip of the mouth to the median projection of caudal fin lobes with the EventMeasure software (SeaGIS Pty Ltd. Bacchus Marsh, Victoria, Australia; Fig. S1B).

We took special attention during video processing to minimize the possibility of counting fish twice. Whenever two similar fish were observed and there was a doubt on whether it was the same individual, length estimated by the software was examined. If sizes were very different and the accuracy index (RMS; SeaGIS Pty Ltd. Bacchus Marsh, Victoria, Australia) indicated by the software was acceptable (<20 mm), fish were considered to be different individuals. Further, additional features in the coloration and distinct marks on the individuals were considered.

## Statistical analysis

### Total length data

Considering the hierarchical structure of our data with individual body size per species nested within each video station and each video station nested within each locality, we used Generalized Additive Mixed Effects Models (GAMMs) fit by REML (statistical packages "nlme" v. 3.1–157 and "mgcv" v. 1.8–39) (*Barry & Welsh, 2002*). The Mixed Effects Models approach allowed us to examine the overall interaction between species, sex, years, and localities while accounting for within study variance structure due to random effects. In our case, video-stations were used as random effect due to the expected correlation between the multiple observations (individual body size) from the same video station.

For the optimal model selection, we followed the top-down protocol described by *Zuur et al. (2009)*. Therefore, we started with a full linear regression model, including every possible explanatory variable within the fixed part (sex, species, localities, year) and their interactions. To include video stations, we fit the model with generalized least-squares (GLS) regression models and compare the linear regression model with the linear mixed effect model (LMEM) using the likelihood ratio test and confirmed with the Akaike Information Criterion (AIC; Table S2) (*Burnham & Anderson, 2003*). Finally, we performed model validation by inspecting the residuals of the best-fitted models (statistical package "lattice" v. 3.1–157), as patterns were observed within the residuals, we applied GAMMs fitted by REML. Species without sex designation were excluded when the sex explanatory variable was included as fixed effect in the model. Likewise, BOL and ALB were excluded from the model when comparing data for 2018 and 2019, as only SA and PRO were surveyed during those years (Table S2).

### Abundances and biomass

Biomass was calculated per individual with the estimated individual total length and the equation B = aL$^b$ (*Marks & Klomp, 2003*) where "B" is the biomass in grams, "L" is body length in cm and parameters "a" and "b" are constants available in FishBase (www.fishbase.org). We performed a square root transformation of abundances and biomass data so that each taxon could contribute to the similarity among samples by softening the influence of the more dominant taxa. In addition, we standardized the biomass data by dividing the estimated values per species by the total biomass of parrotfishes at each station per locality. To determine if localities were statistically distinct from each other in their parrotfish community structure, under the fishing gradient established, we ran a PERMANOVA (*Anderson, 2001*) with a Bray-Curtis similarity matrix. Finally, we ran a SIMPER test (*Clarke, 1993*) to assess the average percent contribution of each species to the dissimilarity (statistical packages "vegan" v. 2.5–7; "ggplot2" v. 3.3.5).

All analyses used a *p*-value significance level of 5% and were carried out in R (v 4.0.2, R Foundation for Statistical Computing, Vienna, Austria). In addition, the data gathered in 2018 and 2019 were analyzed independently as we did not sample BOL, and ALB in the second year. Likewise, to evaluate changes in time, only the data from SA and PRO in 2018 were included in the analysis.

## RESULTS

We recorded a total of 10,438 parrotfishes (6,665 in 2018 and 3,763 in 2019). The most frequently observed species were *S. iseri*, *S. taeniopterus*, *S. aurofrenatum*, and *S. viride*. As expected, the large-sized species *S. coelestinus*, *S. coeruleus*, and *S. guacamaia* were rare and almost exclusively observed in PRO, the site with the lowest level of fishing pressure. We were able to estimate the total length of 9,788 individuals, corresponding to over 90% of the reported occurrences. Abundance, biomass, and individual body size of parrotfishes varied widely among the islands, species, and sexes (Tables S3 and S4). We present the results per metrics evaluated, emphasizing the general patterns observed per size category under the fishing pressure gradient proposed.

### Variation of the individual total length

The best-fitted models for the individual total length had sex, year, species and locality as fixed effects, including the interaction between them and video stations as random effects (Table S2).

From the 2018 data set, the total length of the medium-sized species showed a negative relationship with fishing pressure, with the smallest individuals found on the most heavily fished island (SA, Table S3; Fig. 2A). *Sparisoma viride* had larger sizes in PRO (β = 4.77, SE = 8.77, *p* = 1.91e−06; Fig. 2A) and significantly smaller ones in SA (β = −5.39, SE = 9.42, *p* = 7.25e−08; Fig. 2A). Females of the species reached larger sizes in PRO (β = 5.81, SE = 12.84, *p* = 0.000275; Fig. 2C), and smaller in SA (β = −4.47, SE = 8.95, *p* = 8.14e−06; Fig. 2C) while males had greater sizes in BOL (β = 8.45, SE = 3.86, *p* = 0.00011; Fig. 2B) and smaller in SA (β = −5.98, SE = 11.02, *p* = 3.11e−09; Fig. 2B). On the other hand, *S. vetula* had larger sizes in PRO (β = 3.64, SE = 13.01, *p* = 6.7 e−09; Fig. 2A) and BOL (β = 3.26,

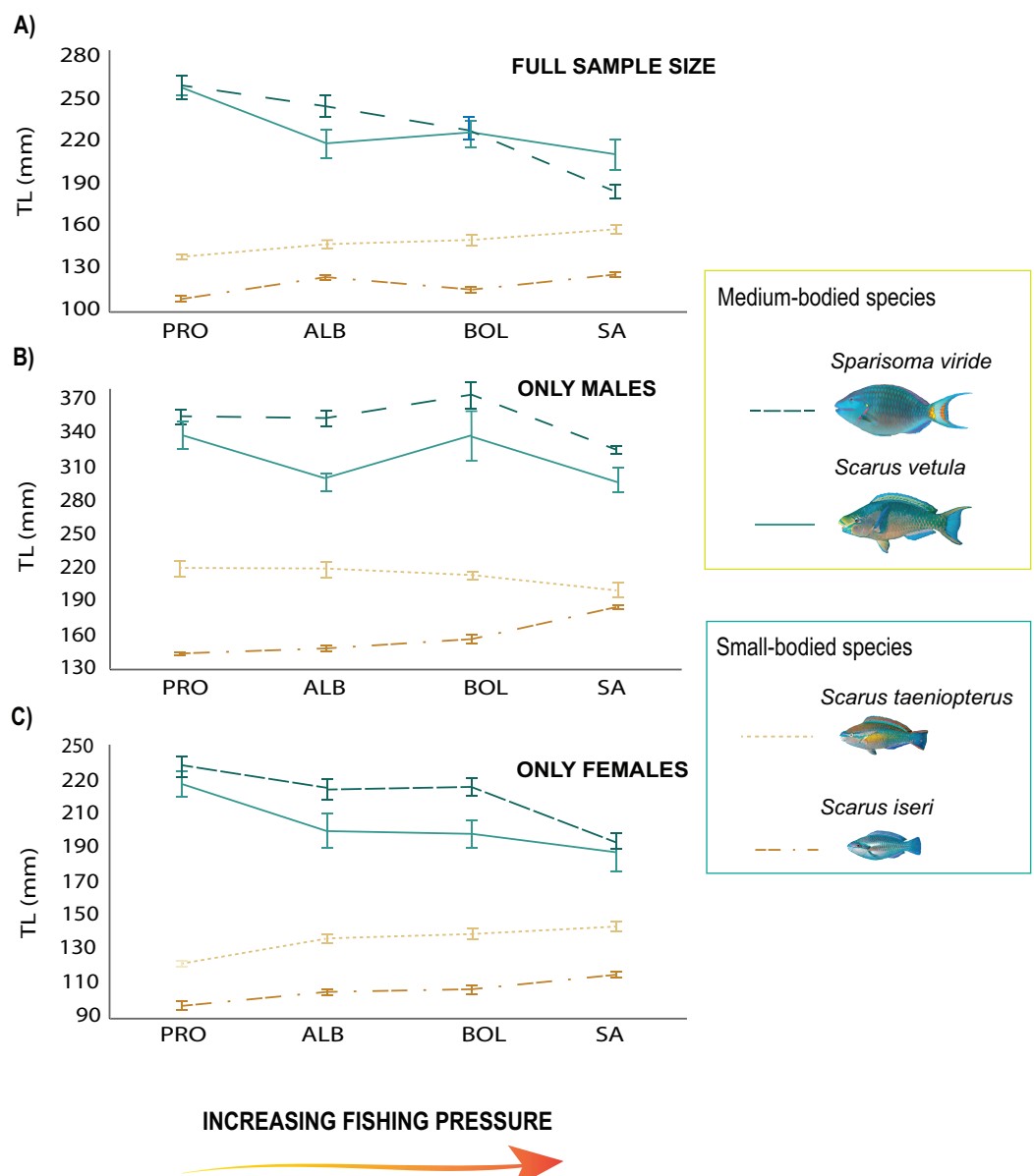

**Figure 2 Trends in mean body size of parrotfish species under the fishing pressure gradient.** (A) Variogram with standard error (SE) bars on independent means of body size of the full sample size of the species populations in 2018, (B) variogram with standard error (SE) bars on independent means of body size of males, (C) variogram with standard error (SE) bars on independent means of body size of females. Species trends are specified with line shapes and their size category are specified with colors.

SE = 12.34, $p$ = 0.0011; Fig. 2A), with females reaching larger sizes in the former ($\beta$ = 5.81, SE = 12.84, $p$ = 0.000275; Fig. 2C). In contrast, only the males of the small sized species *S. iseri* had larger individual body sizes at the most heavily exploited island of SA ($\beta$ = 3.1, SE = 7.6, $p$ = 0.0019; Fig. 2C). *Sparisoma aurofrenatum* showed no clear pattern related to fishing pressure, although their individual body sizes presented significant differences among the localities ($\beta$ = 3.8, SE = 7.34, $p$ = 0.00014).

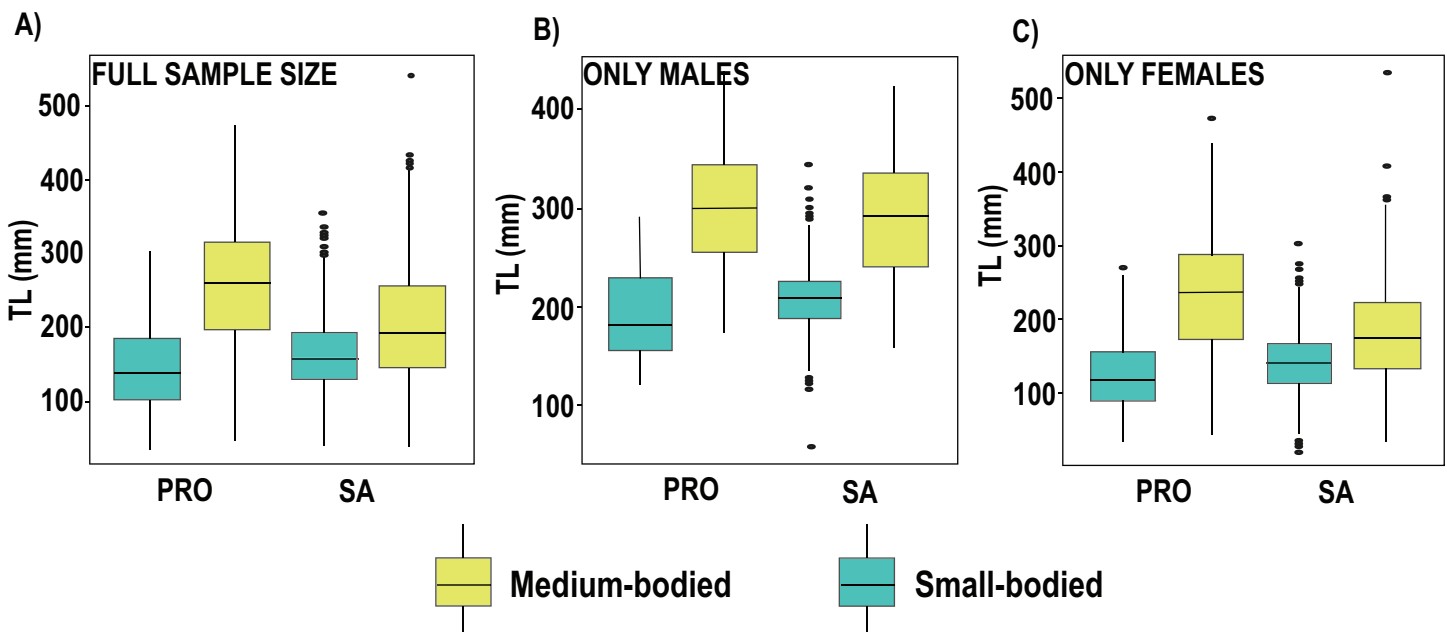

**Figure 3 Boxplot of total length (TL) of parrotfish species collected in 2019 in San Andrés (SA) and Providencia (PRO).** (A) Boxplot of total length calculated of the full sample size of medium and samall-bodied species populations, (B) boxplot of total length calculated for males of medium and samallbodied species populations, (C) boxplot of total length calculated for females of medium and samall-bodied species populations. Medium-bodied species include: *Sparisoma virde, Scarus vetula, S. rubripinne* and *S. chrysopterum*. Small-bodied species include: *S. taeniopterus, S. aurofrenatum, S.iseri* and *S. atomarium*.

We found no differences between the 2 years evaluated (DF = 6, F = 1.45, *p* = 0.19). Nevertheless, in 2019 the differences between localities and species remained the same, with the medium size species showing smaller sizes in SA, including *S. vetula* (β = −6.01, SE = 8.91, *p* = 2.13s−09; Fig. 3; Table S4) and *S. viride* (β = −10.3, SE = 7.53, *p* = <2e−16), and the small-sized *S. iseri* having larger sizes in SA (β = 3.73, SE = 6.68, *p* = 0.0002). In addition, in 2019 we were able to gather a greater sample size of the other two medium-bodied species, and both species showed smaller sizes in SA: *S. rubripinne* (β = −5.78, SE = 12.01, *p* = 8.27e−09) and *S. chrysopterum* (β = −3.49, SE = 10.57, *p* = 0.00049).

## Variation of abundances and biomass

The PERMANOVA applied to parrotfishes abundance in 2018 showed significant differences (DF = 3, MS = 1013.2, *F* = 1.99, *p* = 0.01) between SA and the remaining three localities (DF = 30; SA:ALB *t* = 1.59, *p* = 0.023; SA:BOL *t* = 1.52 *p* = 0.026; SA:PRO *t* = 1.71, *p* = 0.026). The results dropped by the SIMPER test showed that the greater abundances of *S. taeniopterus* females in SA contributed the most to the differences between this island and the other three localities (Fig. 4; Table S5). We also found significant differences in 2019 between SA and PRO in the proportions of the parrotfish composition (DF = 1, MS = 3849.5, *F* = 4.44, *p* = 0.002). Likewise, *S. taeniopterus* females showed to be more abundant in SA and were largely the major contributors to the differences found by the SIMPER analysis (Table S6).

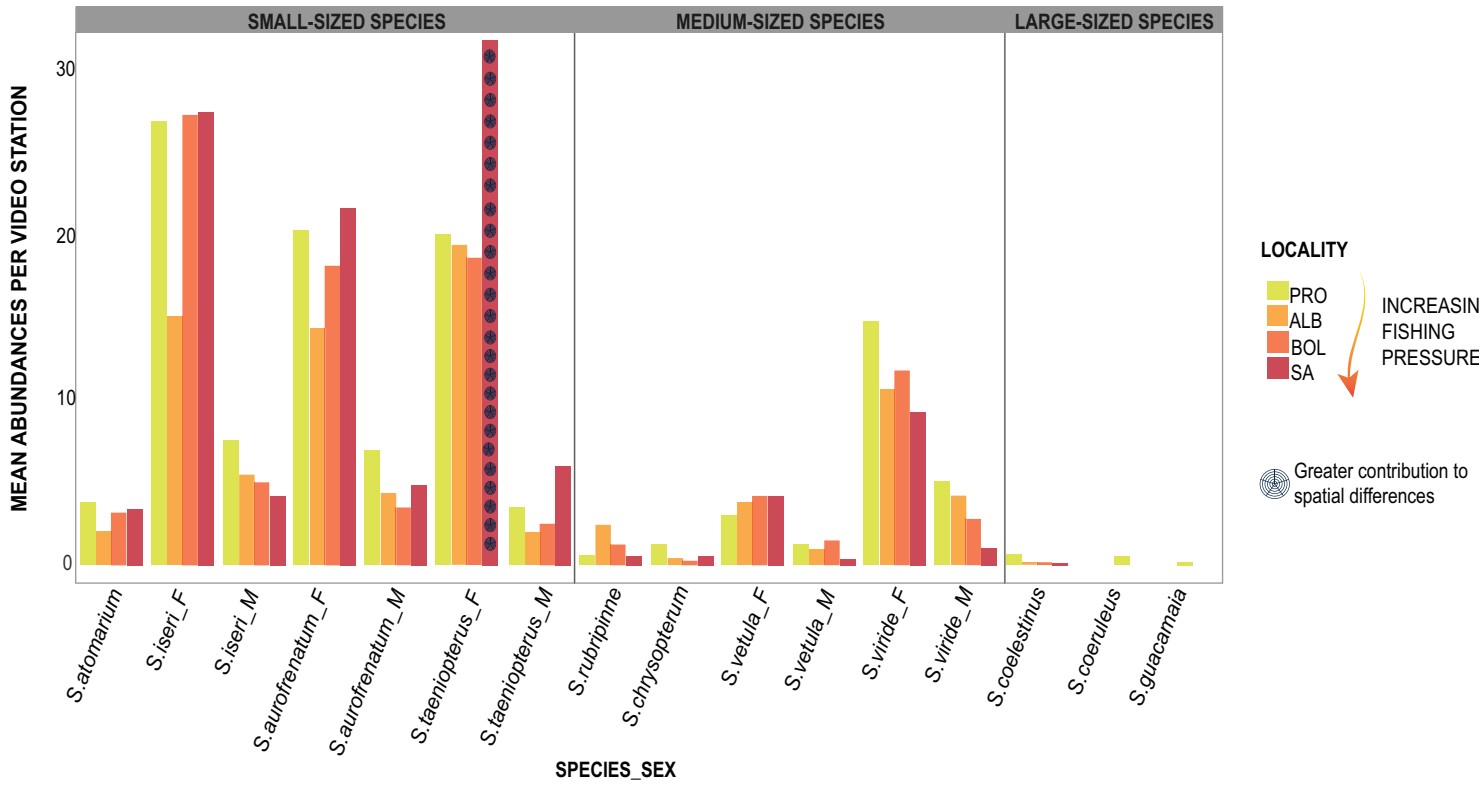

**Figure 4 Parrotfishes occurrences.** Bars represent the mean abundance found per video station of each species at the four oceanic islands. The species are grouped into the size categories established and differentiated between males (_M) and females (_F).

Biomass data showed significant differences (DF = 3, MS = 1903.1, $F$ = 2.81, $p$ = 0.01) between SA and the other three localities (DF = 30; SA:ALB $t$ = 2.28, $p$ = 0.003; SA:BOL $t$ = 1.92 $p$ = 0.012; SA:PRO $t$ = 2.39, $p$ = 0.002). Also, the 2019 biomass data showed SA and PRO to be significantly different (DF = 1, MS = 4045.2, $F$ = 4.57, $p$ = 0.004). In both studied years, the species that contributed most to the overall differences were the small-sized species *S. taeniopterus*, and the medium-sized species *S. vetula* and *S. viride* (Tables S7 and S8). The former had a higher load in the overall parrotfish biomass in SA, while the other two contributed in greater proportion in the other three localities. In addition, at the least exploited site of PRO we found a higher contribution of the medium-bodied species *S. rubripinne* and *S. chrysopterum* and the large-bodied species *S. coelestinus*, *S. coeruleus*, and *S. guacamaia* (Figs 5; Tables S6 and S7). As with individual body sizes, no significant differences were found at SA and PRO between the 2 years evaluated in terms of abundances (DF = 1, MS = 929.32, $F$ = 1.57, $p$ = 0.158) and biomass (DF = 1, MS = 851.61, $F$ = 0.9, $p$ = 0.474).

## DISCUSSION

Effects of artisanal fishing pressure on parrotfish species populations, sex change, and sex ratios have been previously described (*Hawkins & Roberts, 2003*, *2004*; *Hardt, 2009*; *Vallès & Oxenford, 2014*; *Vallès, Gill & Oxenford, 2015*; *O'Farrell et al., 2016*). In this study, we
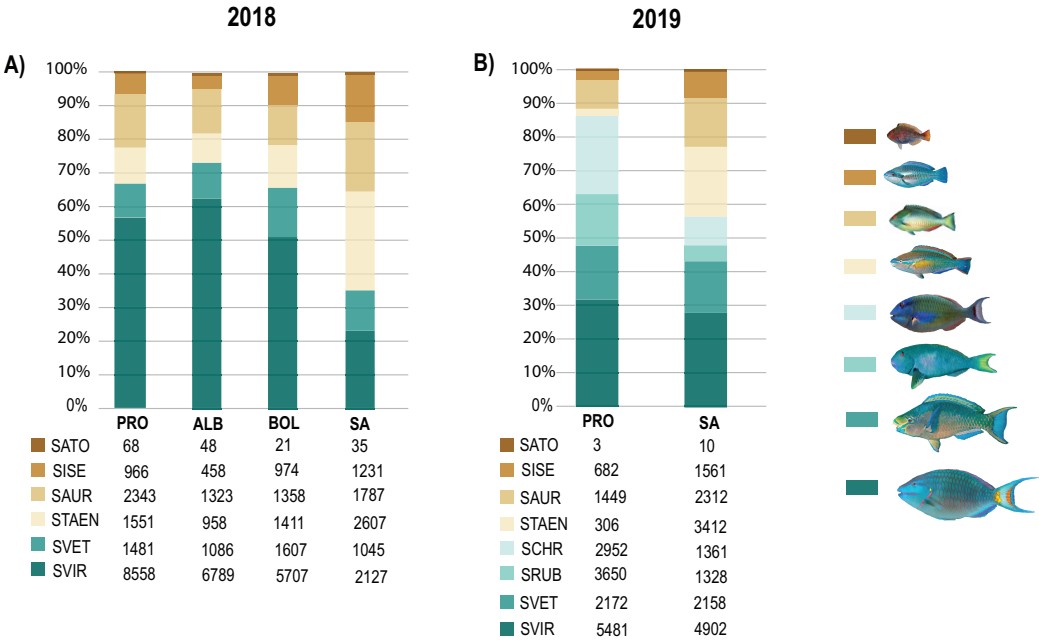

**Figure 5 Species contribution to the overall parrotfish biomass.** (A) Percentage of contribution of the mean biomass per species over a gradient of fishing pressure: San Andres (SA), Bolívar (BOL), Albuquerque (ALB), and Providencia and Santa Catalina (PRO), (B) Percentage of contribution of the mean biomass per species in SA and PRO. Only species with over 30 individuals per locality are included: Medium-bodied species: *Sparisoma viride* (SVIR), *S. rubripinne* (SRUB) and *S. vetula* (SVET). Small-bodied species: *S. taeniopterus* (STAEN), *S. iseri* (SISE), *S. aurofrenatum* (SAUR), and *S. atomarium* (SATO). Inserted tables show the mean biomass (g) per specie at each locality.

have further investigated these relationships under a qualitative fishing pressure gradient, including a comprehensive sample size of eleven species and a high-resolution technique for the underwater assessment of body sizes (diver-operated stereo-video). Overall, our data suggest that fishing pressure is indeed related to the spatial variation in abundance, biomass, and individual body size of the parrotfish species, but most importantly, that there might be indirect effects of fishing pressure on small-sized species.

Following our initial predictions, we expected to find smaller sizes and lower proportions of large and medium-bodied species at a greater fishing pressure. This was indeed the case for most of the species and is consistent with previous studies, where a negative relationship between fishing pressure, body sizes, sex ratios, and abundances were found (*Hawkins & Roberts, 2003*, *2004*; *Vallès & Oxenford, 2014*; *Vallès, Gill & Oxenford, 2015*). Likewise, Mixed Effects Models considering sex showed that females of the medium-bodied species *S. vetula* and *S. viride* had larger sizes at lower fishing pressure. This finding could be due to selective fishing for large females at the high fishing pressure sites, but it could also indicate that, at least in less exploited locations, sex change occurs at larger sizes than at more heavily fished grounds (*Hawkins & Roberts, 2004*; *O'Farrell et al., 2016*). What was most striking, however, were the differences found in the small-sized species *S. iseri* and *S. taeniopterus*. We found that parrotfish proportions at the most heavily fished locality showed significant differences with the other three least exploited

fishing grounds evaluated and that higher abundances and biomass of *S. iseri* and *S. taeniopterus* contributed the most to those differences. Furthermost, this is the first documented study, to our knowledge, to record larger individual sizes of a small-sized parrotfish species at a heavily fished locality, as is the case of the males of *S. iseri* in SA.

Differences found in biomass are certainly driven by the mentioned changes observed in the abundances and individual body sizes. The higher proportion of small-sized species in the total assemblage of parrotfishes predicted at the beginning of this work is not only a product of a lower contribution of individual body size and proportions of medium and large-bodied species but also an outcome of the higher abundances and individual body sizes of small species. We consider that the higher contribution of *S. iseri* and *S. taeniopterus* in the most heavily fished locality suggests an indirect effect of fishing on these two small-sized parrotfish species. *Dulvy et al. (2004)* suggested that indirect effects of fishing pressure may be due to reduced predation or interspecific competition. In theory, once predators are removed from the ecosystem, their preys should increase as there is no control over them, yet, there is no clear evidence of this relation (*Clua & Legendre, 2008*). Some studies have found no changes in preys after the demise of predators (*Jennings & Polunin, 1997*; *Taylor et al., 2018*; *Roff et al., 2019*). Other authors have linked greater biomass of predators such as *Epinephelus striatus* with smaller sizes of *S. iseri* and *S. aurofrenatum* (*Mumby et al., 2006*). In our study, although the greater size and abundance of *S. iseri* and *S. taeniopterus* due to lower predation in SA cannot be completely ruled out and requires further examination, to our knowledge, there are no potential predators that may be significantly more abundant in BOL, ALB, and PRO in contrast to SA. Also, as described by *Mumby et al. (2006)*, we should have found greater sizes of the other small-sized species (*i.e.*, *S. aurofrenatum*, which is smaller than *S. taeniopterus*) at the locality with supposedly fewer predators due to higher fishing pressure (SA); still, this was not the case.

Instead, we believe that competitive release might be playing a role in the results found with *S. iseri* and *S. taeniopterus*. Through stable isotope analysis and intestinal content, *Dromard et al. (2015)* recognized an overlap in the trophic niche between *S. vetula*, *S. viride*, *S. iseri* and *S. taeniopterus*. Therefore, it could be argued that when the medium-bodied species (*i.e.*, *S. vetula*, *S. viride*) are removed from the system, the small-bodied ones (*i.e.*, *S. iseri*, *S. taeniopterus*) may be favored, as some resources for which they actively compete become more available. Consequently, in an intensely exploited locality such as SA, small species can reach larger sizes and higher abundances than in less exploited localities, where larger species limit the resources they compete for. In addition, *Hawkins & Roberts (2004)* found greater abundances of the smallest species of parrotfish (*S. atomarium*) in Jamaica, the most heavily exploited island in their study. The intense pressure exerted on parrotfishes in Jamaica has been documented (*Hardt, 2009*), and we wonder if SA, the most heavily exploited island considered in our study, is showing a step in a chain of changes that follows as parrotfishes are gradually depleted according to sizes of interest for fisheries. Hence, it is essential to monitor parrotfish populations in the Reserve as small size species, like *S. aurofrenatum*, may become of interest to fisheries. Such monitoring may also give insights into phase shift processes in
coral reefs as *S. aurofrenatum* is a critical consumer of the seaweed *Dyctiota* in the Caribbean reefs (*Dell et al., 2020*), a highly opportunistic algae and a great competitor of corals (*Fong et al., 2003*).

The differences found in parrotfish assemblages can likely influence many biological traits of the species, such as reproduction, but also at the ecosystem level. On the one hand, given the positive relationship between body length and egg production, the reduction of mean body sizes of parrotfish populations due to overfishing is probably negatively affecting their reproductive productivity (*Hawkins & Roberts, 2004*). On the other hand, the differences found in this study could have a flow-on influence on ecosystem functions (*Bonaldo & Bellwood, 2008*; *Robinson et al., 2019*; *Shantz, Ladd & Burkepile, 2020*). Larger individuals of parrotfishes have greater roles in processes like bioerosion, corallivory, and grazing of certain macroalgae (*Bonaldo, Hoey & Bellwood, 2014*). *Shantz, Ladd & Burkepile (2020)* found that the removal of large parrotfish on Caribbean coral reefs increases the abundance of macroalgae and decreases the growth of massive and reef-building corals. This result suggests that it should be necessary to protect fishes larger than 20 cm to maintain their functional diversity. Hence, the functional roles of large parrotfish species are not necessarily replaced by higher abundances or larger sizes of the small-sized species. Bioerosion and coral predation are two ecological functions highly susceptible to overfishing as their rates increases with the body size of parrotfishes (*Bellwood, Hoey & Hughes, 2012*). *Scarus vetula* and *S. viride* are important bioeroders (*Bruggemann et al., 1996*), and they had smaller sizes in SA; this could cause lower bioerosion rates at the locality. Concerning the consumption of algae, recent studies show that species not consistently considered as part of the herbivore community in the Caribbean, such as *Acanthurus coeruleus, Acanthurus tractus*, and *Kyphosus* spp., are critical consumers of macroalgae (*Duran et al., 2019*; *Dell et al., 2020*), which are the type of algae that are dominating degrading coral reefs (*Perry et al., 2018*). Consequently, future studies should include these groups of herbivores to determine the effect of herbivores removal on the ecosystem function.

Our findings did not show significant differences between the 2 years evaluated. However, sampling efforts differ between years as a greater number of shallower stations nearby seagrass meadows were examined in both localities (SA and PRO) in 2019. Therefore, it is relevant to continue monitoring the parrotfish populations, hopefully at the same stations to accurately evaluate possible differences in time. Nonetheless, our sampling effort was balanced between the localities in 2019 and allowed us to gather a greater sampling size of *S. rubripinne* and *S. chrysopterum*. Those two medium-sized species had smaller sizes in the most heavily fished locality. At present, neither of these two species has a threat category assigned in Colombia (*Chasqui et al., 2017*), and it is premature to speak of threat due to fishing pressure when their comparable data is only given in two localities. However, it is important to consider that *S. chrysopterum* had a negative relationship with fishing pressure (*Vallès & Oxenford, 2014*) and, according to a report from the continental shores of Colombia, it is one of the most captured species of parrotfishes (*INVEMAR-MADS, 2017*). Therefore, we recommend increasing the sampling

effort by focusing on gathering data of both species on a temporal and spatial level to better understand the differences in our study.

Finally, among the study limitations, different factors were not considered that may influence fish herbivore species' body size and abundances. Such items may include habitat quality, food availability, and other environmental variables. In some studies, factors such as productivity, temperature, rugosity, latitude, geomorphological attributes of reef formations and slope have shown weak relationships with parrotfish community changes (*Cinner et al., 2013*; *Taylor, 2014*; *Taylor et al., 2018*). In contrast, coral cover and position of coral reefs seem to be related to changes in biomass and abundance of parrotfishes (*Hoey & Bellwood, 2008*; *Nash et al., 2012*). Exposed reefs have greater biomass of parrotfishes and erosion levels (*Russ, 1984*; *Gust, Choat & McCormick, 2001*; *Hoey & Bellwood, 2008*; *Taylor, 2014*; *Taylor et al., 2018*; *Roff et al., 2019*). Our findings show that *S. vetula* and *S. viride* reach larger sizes in the less fished locality (PRO) and in BOL, with males of *S. viride* reaching larger sizes in the former. Larger sizes and levels of bioerosion of both species are associated with exposed reefs (*Koltes, 1993*; *Bruggemann et al., 1996*; *Clements et al., 2017*), and BOL seem to have more exposed environments than the other localities (*Gamboa et al., 2012*) which could explain why such differences were found in this site. Consequently, differences in body size are explained by different factors such as reefs' exposure, resource availability, and demographic characteristics of fish stocks influenced by external stressors such as fishing exploitation. Future efforts should include environmental factors such as reef exposure and coral cover to account for our study's intraspecific variation in body size.

## CONCLUSIONS

Abundances, individual body sizes and biomass of parrotfish species showed significant differences and varied widely depending on the fishing pressure gradient and the size category of the species. On the one hand, medium and large-bodied species of parrotfishes had a negative relation between fishing pressure and contribution to the overall biomass, occurrences, and individual body sizes. On the other hand, we found a significant contribution of the small-size species evaluated to the overall biomass at higher fishing pressure sites. The such greater contribution was driven by the reduction of sizes and abundances of medium and large-bodied species, directly affected by fisheries but was also a product of greater abundances and, most interestingly, larger sizes of the small-sized species. We suggest this result points to indirect effects of overfishing and that such effects could related to a possible case of competitive release. It is relevant to establish how such variations are linked with possible effects on the species' reproductive productivity and ecosystem processes.

Overall this study provides insights into intraspecific variation in body size, and our substantial volume of data adds to the existing literature on the extent of variation in parrotfish assemblages over relatively short distances. While we consider our general results are better explained under a fishing pressure perspective, this by no means denies the influence of other factors such as inter-island environmental differences. Therefore, although the direct and indirect effects of fishing pressure suggested by our work deserve

attention, they should be taken as preliminary until further examination on a temporal and spatial scale is conducted.

## ACKNOWLEDGEMENTS

We would like to extend our sincere gratitude to the Comisión Colombiana del Océano (CCO) for organizing and leading the expeditions to Isla Cayo Albuquerque in 2018 and to Providencia and Santa Catalina in 2019 which allowed us to gather part of the data used in this research. We also thank A. Puentes and J. Carvajal for their research assistance. The assistance and support of Martha Patricia Rincón-Díaz (National Patagonian Center - CONICET) and James Robinson (Lancaster University) is greatly appreciated. Contribution No. 544 of CECIMAR, Instituto para el Estudio de las Ciencias del Mar of the Universidad Nacional de Colombia Caribbean campus.

### Funding

This research was funded by a Small Grant Rufford Foundation (Grant Agreement No. 25323-1) and by the Science, Technology and Innovation Department- Colciencias. The funders had no role in study design, data collection and analysis, decision to publish, or preparation of the manuscript.

### Grant Disclosures

The following grant information was disclosed by the authors:
Small Grant Rufford Foundation: 25323-1.
Science, Technology and Innovation Department-Colciencias.

### Competing Interests

The authors declare that they have no competing interests.

### Author Contributions

- Natalia Rivas conceived and designed the experiments, performed the experiments, analyzed the data, prepared figures and/or tables, authored or reviewed drafts of the article, and approved the final draft.
- Arturo Acero P. conceived and designed the experiments, performed the experiments, authored or reviewed drafts of the article, and approved the final draft.
- José Tavera performed the experiments, authored or reviewed drafts of the article, and approved the final draft.

### Data Availability

 Raw measurements are provided in the RAW file sheet 1. The abundances per species and locality are provided in seet 2. This is the data used for the statisical analysis to compare among localities.

## Supplemental Information

Supplemental information for this article can be found online at http://dx.doi.org/10.7717/peerj.14178#supplemental-information.

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
