# Peer review of "Spatial variation of parrotfish assemblages at oceanic islands in the western Caribbean: evidence of indirect effects of fishing?"

_PeerJ, doi:10.7717/peerj.14178_

## Round 0.1 · original submission · Major Revisions

Both the reviewers appreciated the work you've put into this paper, and have offered a number of suggestions here that will improve the manuscript.

·

Basic reporting

I found the main findings slightly difficult to follow, and noticed several places where ideas and results could have been communicated more clearly. First, the structure and content of the introduction needs revisiting. Each paragraph should build on the information in the preceding paragraph, starting with the overarching research context before zooming in to the research questions and study region. Here the introduction is inverted (beginning with Caribbean, then moving to fishing-size effects), and contains details I felt were mostly related to methods (L60-82).

Second, there were several grammar/spelling errors in the manuscript, and some basic information missing from methods and figures (e.g. Fig 1 y axis label), that will need addressed before publication. In your tables and Figure 5, the comma separators look like they represent decimal points – I’d change this to avoid confusion. You could also round biomass / mean length to the nearest g / mm for ease of reading tables. Please see my minor comments for examples of these issues.

Figures and tables were also ambiguous in places and should be improved - see additional comments below.

Experimental design

Overall the research question is interesting: size-based analyses of reef fish communities have typically examined assemblage-level patterns, and the added resolution of parrotfish sex here adds to the existing research on these topics. Introduction could more clearly explain this knowledge gap (see above).

I'm not familiar with stereo-video methods. How did you ensure that fish were not counted twice? How does CAL software calibrate the video, and is this sufficient to quantify fish lengths accurately?

Several methods sections could be expanded to provide more detail. For example, did you use AIC, LRT and REML for model comparison? Are these comparable? Why is ALB lower fishing intensity than BOL?

Apologies for brevity of response - see additional comments for methods and stats queries.

Validity of the findings

Overall this study provides insights into intraspecific variation in body size, and explores potential for indirect competitive release of smaller-bodied species. The study replication and statistical analyses seem appropriate do me (notwithstanding missing methods details, see additional comments).

However, I don’t completely agree with your assessment that sex change is linked to fishing pressure (L253). Smaller females at high fishing pressure may just reflect that there is fishing selectivity for large females, so the populations you survey at SA have low/rare abundance of large-bodied females. Though fishing-induced sex changes may be possible (and observed in other studies), the primary effect underlying your results is size-selective fishing, so I would think that is your also the leading explanation for differences in female size between islands.

Also, inter-island effects on parrotfish assemblages are likely to be strong, but this was only mentioned at the end of the paper. Habitat composition, coral cover, wave energy, algal cover are all known to influence parrotfish biomass and size (Hoey & Bellwood 2008, Coral Reefs; Kirsty Nash’s papers), are you able to comment on how these influences might relate to your own results?

Additional comments

L55 – here and lots of places in the results/discussion, you describe ‘significant relationships’. This could be phrased more clearly by describing the relationship in words, for example, ‘A largescale study by…found that abundance of small-bodied Sp. atomarium increased with fishing pressure’.

L65 – ‘biased interest’ is the same as selectivity for large fish?

L69 – 82 – these are all methodological details, and most are repeated in L98-116

L97 – I assume ALB < BOL fishing intensity because it is further from SA? This wasn’t clear in the methods.

L112 – CPUE = catch per unit effort

L113 – ‘shown a negative trend’ could be replaced with ‘declined’

L128 – I’m not familiar with stereo-video methods. How do you ensure that fish aren’t counted twice?

L155 – what are the random effects terms and why did you choose those?

L158-173 – several statistical tests used here that were not explained. What is AIC, LRT and REML tests, and are they all used for the same purpose? What is dissimilarity?

L191 – in the methods you designated ALB as lower fishing pressure than BOL.

L230 – PRO is the ‘least exploited site’ ?

L268 – predation release could also be occurring, see Mumby et al. 2006 (Science) for predation escape by large parrotfish, meaning predation pressure at lightly fished reefs may be high for small/medium sized parrotfish.

L281-285 – I would avoid vague and speculative statements like this. Are these fishing gears able to target small parrotfish? What is population resilience to fishing? What is total fish community abundance, and habitat condition? All these factors inform fishery resilience, not just the proportion of small fish.

L300 – good interpretation, but missing step for me is that you don’t explain which functions might degrade if large-bodied parrotfishes are depleted, and the implication of this for reef ecosystems. Bellwood et al. 2012 (Proc R Soc B) good context for exploring these ideas more fully.

L317 – this inter-island caveat is important, but shouldn’t be left to the last sentence of the conclusion. What island effects might drive variation in parrotfish populations? Helpful if you can expand on this in a separate Discussion paragraph.

Fig 2 – y-axis label missing. I appreciate the colour palette is designed to match parrotfish colouring, but this figure would be improved with more colours than have greater contrast, so the lines are easily matched to different species (you could use Fig 5 palette for consistency)

Fig 4 – the fishing pressure gradient is reversed compared to Fig 2 (e.g. left to right)

Fig 5 – commas are decimals? What units?

I am signing my review for transparency and invite the authors to contact me if they have any questions.

James Robinson

Reviewer 2 ·

Basic reporting

The manuscript was written in professional English; however, it requires some grammar checks, avoidance of passive voice and repetitive connectors within the manuscript. I made specific comments on these topics that might help with this part. Also, authors may want to pay attention to differences in font type and sizes in the text.

The introduction and background showed relevant information on the general research question, and the literature is well referenced. The structure of the manuscript, supporting materials, and raw data supplied conform to PeerJ standards. Authors may want to provide supporting materials that describe the methods and results of the video calibration by considering that this is a new technique to monitor reef fish species in Colombia. This information would help other local and regional research groups to replicate this technique by allowing comparisons with this study's findings.

Figures are relevant, high quality, and well described. However, I made minor comments on figures 2, 3, 4, and 5 to improve their auto interpretation. As a suggestion: the results' interpretation and connection between the manuscript, graphs, and supplementary material would be straightforward if using the species scientific or the common name, but not both. Furthermore, authors could standardise it, preventing readers from going back and forth throughout the article to connect the results among the three sections.

Experimental design

The primary research meets the scope of the journal. The research question was well defined and relevant to the context of overfishing of large-bodied species in the Caribbean region. The study is also novel within the Colombian scientific research by relating fishing pressure gradients with changes in body size and abundance of fish species used by local markets.

Despite the lack of quantitative information on the local fishing effort, I find it valid that the authors intended to describe a fishing pressure gradient qualitatively by considering scientific criteria and the best available local information. Also, the study was performed to a high technical standard to diminish biases in survey errors and had a good survey size (stations) and several fish individuals with their length measured. This paper's methodological strengths filled some knowledge gaps in the country's ecosystem-based fisheries management, such as the description of fishing pressures on marine fish communities. Also, it is relevant to mention that the study introduced a novel technology for surveying reef fish species in Colombia already used in other geographic areas.

Despite these strengths, the methodology does not provide information regarding the survey site selection based on depth and substrate type. Authors may consider providing this information to show that survey sites were similar in these habitat features, accounting for a balanced design among depths and substrate types and any influence of habitat structure on the length and abundance of parrotfish species. Also, the authors may want to provide a table that shows the procedure and results for model and random effects selection (e.g., AIC ) and the effect size of fixed effects. This table would improve the rationale behind the protocol used to select the final model by helping replicate their methods.

Validity of the findings

The authors discussed the impact of results and encouraged replication and improvements of their methods. The result interpretation indicating local overfishing of large herbivores is in line with indirect effects of competitive release, possible effects on the ecosystem functioning of coral reefs and a vital call to protect large-sized herbivore species in Colombia. Although this interpretation is meaningful, recent studies also highlighted that the functional role of parrotfish does not necessarily contribute to the removal of macroalgae species that overgrow and compete for space with corals, thus changing the paradigm that parrotfishes have a significant influence on the recovery of reef systems that faced phase-shifts (Dell et al. 2020: https://doi.org/10.3389/fmars.2020.00676; Duran et al. 2019: https://doi.org/10.1111/maec.12561). This fact makes questioning if the authors' discussion about ecosystem effects due to parrotfish overfishing needs to be downplayed a little and if a call to include other local herbivore species in these kinds of studies is needed. Authors may also want to discuss the importance of S. aurofrenatum as a critical consumer of Dyctiota algae in the Caribbean reefs (Dee et al., 2020). In addition, the discussion section deserves reasoning about the study limitations due to factors not considered that may influence fish herbivore species' body size and abundances, such as habitat quality and food availability. Including these topics enriches the discussion, and the paper will offer guidance to improve this kind of study.

The authors provided all raw data on surveyed species and robust statistical analyses. However, it would be of great help to provide a table in supplementary material with the outputs of selected models and the effect size of fixed factors for a more straightforward interpretation of the results.

Additional comments

The study used a diver-operated stereo-video to examine individual body size, sex ratios and proportion of species of the parrotfish assemblage and analyze it concerning a qualitative fishing pressure gradient at four oceanic islands in the Colombian Caribbean. Study findings highlight the spatial variation in body length and abundances of the parrotfish species and community structure at relatively short distances and presented new insights on the responses of parrotfish species on a spectrum of body sizes along a gradient of human pressure.

In general, the research question was well defined and is relevant within the context of overfishing of large-bodied species in the Caribbean region. Furthermore, the study related fishing pressure gradients with changes in body size and abundance of fish species used by local markets, which makes the work novel within the Colombian scientific research. Also, the study introduced stereo-video surveys to monitor reef fish species, making the work attractive for Colombia.

I did enjoy reviewing the research described in the paper and provide some specific comments that hope improve the quality of the manuscript. In conclusion, I consider that primary research meets the journal's scope.

Specific comments:

L14 - Please be specific since it is unclear which characterization type is essential to conduct in parrotfish populations.
L21-22: The phrase sounds redundant. Consider changing it to "observed differences are size-category dependent throughout the qualitative fishing pressure."
L39- The statement requires the cite/information source for records of fishing and supply of parrotfishes to local restaurants. If this is a personal opinion of the authors, it should also be specified.
L54- Consider replacing "behavior" with "effect" since the whole paragraph focuses on the effects of fishing on fish communities.
L54-56- Did the authors mean a "high" instead of "wide" fishing pressure?. Getting a positive relationship between a "wide gradient" and body size is confusing. Also, the authors may consider removing the term "gradient" from the line or writing the sentence clearer.
L74- please review the font for "of" since it seems different from the rest of the text.
L100- consider including "that" between the parenthesis and "in" to improve the syntaxes of the sentence.
L103- Consider replacing "fishermen" with "fishers" to improve your manuscript's inclusive language in sciences. Also, replace "of San Andres" with "from San Andrés" to indicate a place of origin.
L105- Consider replacing "independent fishermen who have not been quantified and target many species "with "independent fishers, whose population not characterized and quantified, target multiple species".
L106- consider replacing "many" with "multiple."
L121- This line seems wordy. Please consider replacing "As there were no previous data of this kind using this technique to compare with "with "In the absence of previous data from this technique to compare".
L121-126: No information was provided about site selection according to depth and substrate type in the main manuscript. Could the authors provide this information and also show that survey sites were similar in habitat features, which account for the influence of habitat structure and quality on the length and abundance of parrotfish species?
L126-128 - I did not understand the connection of this methodological goal with the objectives set for the work. If a goal compares the effectiveness of stereo-video surveys to record rare species concerning other methodologies, then it should be raised in the introduction and developed as such in the work. Please consider restructuring or removing these lines.
L131-134 – The authors explained that they conducted more sampling in areas near seagrasses in SA and PRO "to gather more data of yellowtail and redtail parrotfishes, which had a small, simple size in surveys of 2018". They also stated in lines 125-126 that their methodology included a balanced sampling design and that the substrate type was a factor in selecting sampling sites. Considering these two statements, it does not seem very clear to identify if the design was balanced in the sample size per substrate and depth. Could the authors include information (maybe a table) with the sampling efforts by depth and substrate type and discuss whether there might be a potential effect of increased sampling in one substrate on abundance estimates for the two rare species?
L151- add "s" to "specie"
L1152- Did the authors mean "Generalized additive mixed models (GAMMs)" instead of "additive mixed effect models"?
L150-162 – The methodology for selecting fixed and random effects seems adequate and follows Zur's (2009) protocol. However, a table showing the results for model structure and random effects selection would be needed to improve the rationale behind the protocol and model structure selection.
L169 - Did the authors mean" survey" or "station" instead of "sample"?
L144-173 - Please include citations for statistical analyses.
L199-200 - This statement seems more appropriate for the discussion section.
L207 – Is table S3 a good supporting material for this line about length?. Table S3 shows differences in abundance data by species and sex/species among localities.
Figure 2 – The y axis lacks units for the total length of fish species.
Figure 3- change "samall-bodied" to "small-bodied"
Figure 4 – This figure seems important to show differences in parrotfish community structure among the locations/fishing gradient. However, it is difficult to identify which species contribute most to the spatial difference. Could authors consider using a symbol for species/sex that contributes more to the spatial differences to improve the central message of this result?
L222-223 – Consider moving the statement to the discussion section.
L244- I wonder if "abundance and biomass" or "contribution of species to abundance and biomass" would fit better with the variables measured in the parrotfish species among localities instead of "proportions".
Figure 5 – replace "bioamass" with "biomass"
L269-271 – These lines are wordy. Please consider splitting it into two ideas and rephrasing it: "Yet, we believed that a competitive release might play a part. Through stable isotope analysis and intestinal content, Dromard et al. (2015) recognized an overlap in the trophic niche between S. vetula, S. viride, S. iseri and S. taeniopterus."
L269-271 – These lines are wordy. Please consider splitting it into two ideas and rephrasing it: "Yet, we believed that a competitive release might play a part. Through stable isotope analysis and intestinal content, Dromard et al. (2015) recognized an overlap in the trophic niche between S. vetula, S. viride, S. iseri and S. taeniopterus."
L292-300 – The result interpretation indicating local overfishing of large herbivores seems to be in line with possible effects on the ecosystem functioning of coral reefs and a call to protect large-sized herbivore species in Colombia. Although this interpretation is meaningful, recent studies detailed that the functional role of parrotfish does not necessarily contribute to the removal of macroalgae species that overgrow and compete for space with corals, thus changing the paradigm that parrotfishes have a significant influence on the recovery of reef systems that faced phase-shifts (Dell et al. 2020: https://doi.org/10.3389/fmars.2020.00676; Duran et al. 2019: https://doi.org/10.1111/maec.12561). Instead, species not consistently considered as part of the herbivore community in the Caribbean, such as Acanthurus coeruleus, Acanthurus tractus, and Kyphosus spp, show a significant consumption of these macroalgae species compared to some parrotfish species such as Scarus iserti, S. taeniopterus, S. vetula, and S. viride (Dell et al. 2020: https://doi.org/10.3389/fmars.2020.00676; Duran et al. 2019: https://doi.org/10.1111/maec.12561). Also, despite their presence in the San Andres and Providence areas, the surgeonfish species and Kyphosus spp were not considered in the present study. This fact makes questioning if the authors' discussion about ecosystem effects due to parrotfish overfishing needs to be downplayed a little and if a call to include other local herbivore species in these kinds of studies is needed. Authors may also want to discuss the importance of S. aurofrenatum as a critical consumer of Dyctiota algae in the Caribbean reefs (Dee et al., 2020). In addition, the discussion section deserves reasoning about the study limitations due to factors not considered that may influence fish herbivore species' body size and abundances, such as habitat quality and food availability. Including these topics enriches the discussion, and the paper will offer guidance to improve this kind of study.

---

## Round 0.2 · Minor Revisions

Both reviewers thought this manuscript is substantially improved from the initial submission, and have a few minor comments that would further improve the paper.

·

Basic reporting

The manuscript is much improved, great work. Some few suggestions from me, mostly just to bring in the reviewer responses to the actual manuscript.

The introduction builds well and adds great context - I wondered if there was still room to note the lack of information on fishing effects on sex ratios in reef fishes, as this is a question directly tested in the study.

Some minor spelling errors still need correcting (L84 'mainly'; L363 missing apostrophe on reefs, L383 'explained').

The reviewer responses are mostly clear but some are not included in the manuscript itself. Please make sure that you add the extra detail that reviewers have asked for, if you think it is appropriate. For example, in methods, you can state the random effects are video stations due to autocorrelation, and briefly explain the stereo calibration procedure (see below).

Experimental design

Thank you for the information about stereo video methods, and sources of uncertainty. Some of your reviewer response could be included in the manuscript, for example noting that you used an accuracy index to assess video processing performance (what is a good index value?). I appreciate that the CAL software is highly complex, but you could still try to help the reader understand its main approach - bringing in a sentence or two about the calibration procedure and the RMS.

Validity of the findings

Data, conclusions and interpretations are valid.

Additional comments

L353 - 'poor to none relations' is 'weak relationships'? Rugosity and herbivore biomass generally has a strong relationship in other studies.

Figures 4 and 5 still have fishing gradients going left to right (overfished to less fished, SA -> PRO), whereas Figure 2 is left to right (less fished to overfished, PRO -> SA). Suggest these are made consistent to aid with interpretation.

·

Basic reporting

The manuscript showed a considerable improvement in the structure of the introduction, and the background clearly showed the relevant information to address the general research question. However, with these new changes, grammar checks are needed. Therefore, I made several grammar and style edits as track changes throughout the word document that, hopefully, authors find helpful.

In the introduction, the authors may want to explain why developing this study in the geographic area is essential regarding knowledge gaps or the possibility of comparing the parrotfish's biological responses to a small-scale spatial gradient of fishing pressure in the Caribbean. I felt that the importance of the study site and justification of its selection was missing in the introduction. I suggested the knowledge gap in the manuscript, but the authors can add others reasons.

I appreciate the information provided by the authors about the methods and results of the video calibration and encourage them to include this as part of the supplementary materials. The authors did an excellent job explaining to reviewers how they did the calibration and measured the individuals' length with relatively good-quality pictures. Also, they provided important information on how they accounted for solving issues related to individuals being counted twice. Therefore, it is still important that authors explain and show in a clean way how they managed the data collection process. Moreover, this information positions the work as the first in Colombia, helping other scientists overcome related methodological issues.

Figures also showed an improvement in the standardization of colors, fish species names, and measurement units.

Experimental design

Thanks for providing information regarding the survey site selection based on depth and substrate type and modifications in the methods and discussion section about the study design and limitations. It makes the work more valuable, guiding further studies on this topic!

The authors provided a table in supplementary materials to improve the rationale behind the protocol used to select the final model by showing results for model structure and performance (e.g., AIC). This table was helpful; however, some information is still missing in the main document and supplementary material to clarify the model selection process. For example, it is still unclear if the authors used only GAMMs to relate predictors to response variables. In table S2, the authors reported models and compared AIC values only for generalized least-squares (GLS) regression models without random factors and linear mixed-effect models (LMEM) with random factors, which was not mentioned in the methods section (statistical analysis). Also, the authors omitted GAMMs in the model comparisons, even though they claimed to compare the entire model with different random effects structures using the AIC. Why were GAMMs not included in the model comparisons? The other reviewer made a specific note regarding using other statistical tests that were not explained in the manuscript, and I agree after reviewing table S2. Therefore, in the methods or supplementary material, it is necessary to clarify why the authors used the GLS and LMEM and why the GAMM was not included in the model comparison. Probably that is a simple step that solves this issue. In addition, authors may want to consider justifying and specifying the sources of random variation and show the effect size estimates of each fixed factor in table S2 that are still pending in the manuscript, as well as the random factor variance and models' marginal and conditional r2's in table S2 to inform the influence of fixed and random factors on explaining model variation.

Validity of the findings

The authors did an excellent job expanding the discussion to include other potential ecosystem effects of fishing pressure, such as grazing and study limitations.

Additional comments

I enjoyed reviewing the research described in the paper and provided specific comments in the manuscript to improve the quality of it, however, the system only allowed me to attached a PDF and not the word document.

I still consider that primary research meets the journal's scope. I have decided to provide my contact information for the transparency process of this review in case the authors need some clarifications on my comments, and need the word document with track changes.

Kind regards,

Martha Patricia Rincón Díaz
E-mail: princon7@gmail.com; mrincon@cenpat-conicet.gob.ar

---

## Round 0.3 · accepted · Accept

Thanks for making all the requested changes to the paper, congratulations!